# Genetic enhancement of *Trichoderma asperellum* biocontrol potentials and carbendazim tolerance for chickpea dry root rot disease management

**Ramangouda G.**[1,2], **M. K. Naik**[2], **Rahul B. Nitnavare**[3,4], **Richa Yeshvekar**[5], **Joorie Bhattacharya**[1,6], **Pooja Bhatnagar-Mathur**[1¤], **Mamta Sharma**[1] *

1 International Crops Research Institute for the Semi-Arid Tropics (ICRISAT), Patancheru, Telangana, India, 2 Department of Plant Pathology, University of Agricultural Sciences, Raichur, Karnataka, India, 3 Division of Plant and Crop Sciences, School of Biosciences, University of Nottingham, Sutton Bonington, United Kingdom, 4 Plant Science Department, Rothamsted Research, Harpenden, United Kingdom, 5 Centre for Plant Sciences, School of Biology, University of Leeds, Leeds, United Kingdom, 6 Department of Genetics, Osmania University, Hyderabad, Telangana, India

¤ Current address: International Maize and Wheat Improvement Center (CIMMYT), México-Veracruz, El Batán Km. 45, Mexico

* mamta.sharma@cgiar.org

**Data Availability Statement:** All relevant data are within the paper and its Supporting Information files.

## Abstract

Advances in biocontrol potentials and fungicide resistance are highly desirable for *Trichoderma*. Thus, it is profitable to use mutagenic agents to develop superior strains with enhanced biocontrol properties and fungicide tolerance in *Trichoderma*. This study investigates the N-methyl-n-nitro-N-nitrosoguanidine (NTG) (100 mg/L) induced mutants of *Trichoderma asperellum*. Six NTG (3 each from 1st & 2nd round) induced mutants were developed and evaluated their biocontrol activities and carbendazim tolerance. Among the mutant N2-3, N2-1, N1 and N2-2 gave the best antagonistic and volatile metabolite activities on inhibition of chickpea *F. oxysporum* f. sp. *ciceri*, *B. cinerea* and *R. bataticola* mycelium under in vitro condition. Mutant N2-2 (5626.40 µg/ml) showed the highest $EC_{50}$ value against carbendazim followed by N2-3 (206.36 µg/ml) and N2-1 (16.41 µg/ml); and succeeded to sporulate even at 2000 µg/ml of carbendazim. The biocontrol activity of N2-2 and N2 with half-dose of carbendazim was evaluated on chickpea dry root rot under controlled environment. Disease reduction and progress of the dry root rot was extremely low in $T_7$ (N2-2 + with half-dose of carbendazim) treatment. Further, carbendazim resistant mutants demonstrated mutation in *tub2* gene of β-tubulin family which was suggested through the 37 and 183 residue changes in the superimposed protein structures encoded by *tub2* gene in N2 and N2-2 with WT respectively. This study conclusively implies that the enhanced carbendazim tolerance in N2-2 mutant did not affect the mycoparasitism and plant growth activity of *Trichoderma*. These mutants were as good as the wild-type with respect to all inherent attributes.

**Funding:** a) ICRISAT supported the study b) "The organization/funders had no role in study design, data collection and analysis, decision to publish, or preparation of the manuscript." c) "The authors received no specific funding for this work."

**Competing interests:** The authors have declared that no competing interests exist.

## Introduction

*Trichoderma* spp. are among the most promising biocontrol agents used against numerous plant pathogenic fungi [1]. *Trichoderma* is known to stimulate plant health by protecting the plant *via* mycoparasitism, anti-microbial secondary metabolite production, as well as local and induced systemic resistance against invading pathogens [2]. There are several reports on *Trichoderma* species as biocontrol agent, however, only a handful of species like *T. harzianum*, *T. afroharzianum*, *T. viride*, *T. asperellum*, *T. koningiopsis*, and *T. virens* based formulations are used to manage the plant diseases [1, 3].

Often *Trichoderma* species exhibits greater variability in antagonistic capacity and bio-stimulant action on plant pathogens and plants [4]. Some strains are more suitable for biological control of disease and others for stimulating crop growth and nutrient uptake [5]. The limited efficacy and inconsistent performance against targeted pathogens remains a major drawback in the field conditions [6]. Various strategies have been employed to enhance the efficacy and consistency of biological control activity. Recurrent selection of *Trichoderma* strains, and random mutations with UV light or chemical mutagenesis followed by subsequent selection has been a successful strategy to enhance efficacy and fungicide tolerance of *Trichoderma* species [7].

There have been several successful reports regarding the development of novel *Trichoderma* strains by using mutagenesis which has led to enhanced fungicide tolerance and biocontrol potential [1]. Pesticide tolerance in *Trichoderma* would be a prerequisite to reduce the fungicide resistance in pathogens and to promote the plant health [8]. Pesticide-resistant *Trichoderma* in combination with reduced fungicide application would help in reducing the detrimental effects of the latter on soil along with ensuring satisfactory levels of crop protection [9]. Moreover, reduced amount of chemical usage could weaken pathogen propagules for subsequent attack and give additive/synergistic effect on plant growth [10].

Among fungicides, carbendazim is one of the most extensively used benzimidazole (methyl benzimadazol-2-yl carbamate) for control of soil-borne diseases through seed treatment and foliar application [8]. Carbendazim resistance in pathogen populations have been detected in the field shortly after an intensive and exclusive exposure [11]. However, reversion of resistant populations has not been reported even after the selection pressure was removed [12]. Primarily, benzimidazole functions by inhibiting the polymerization of tubulin monomers into functional microtubules in mitosis and through sequence rearrangement of 5'-flanking region of β-tubulin gene in some species [13, 14]. The benzimidazole tolerance in *Trichoderma* strains are yet to be ascertained for enhanced fungicide tolerance and large-scale field applications. There have been reports to confirm benzimidazole tolerance in *Trichoderma* with substitution of amino acids as a result of single and double mutation in β-tubulin *tub2* region [8, 15].

The aim of this study was to develop *T. asperellum* with enhanced biocontrol activity and carbendazim tolerant mutants of the species by induced mutation followed by characterization of biocontrol potential of mutants. The study also demonstrates dry root rot (DRR) disease control ability with half-dose of carbendazim and its potential in integrated management of soil-borne diseases under field condition.

## Materials and methods

### Origin and maintenance of strains

*Trichoderma asperellum* (WT) (KY575523) strain used in the study was obtained from the University of Agricultural Sciences (UAS), Raichur, Karnataka, India [16, 17]. Three pathogen cultures namely *Fusarium oxysporum* f. sp. *ciceri* (Foc), *Rhizoctonia bataticola* (Rb) and *Botrytis cinerea* (Bc) causing fusarium wilt (FW), dry root rot (DRR) and botrytis grey mould

(BGM) diseases, respectively, in chickpea were obtained from Legumes Pathology, Division of Integrated Crop Management Theme, ICRISAT, Patancheru, India [18].

The cultures were reactivated in potato dextrose agar (PDA) (HiMedia M096) medium and then grown in test-tubes containing 10 ml of the same medium. These cultures were stored in a refrigerator at 4˚C for future use in assays. For each experiment, samples taken from these test-tubes were cultured for five days in Petri dishes (90×15 mm) containing 20 ml of PDA medium.

## Chemical induction and isolation of carbendazim tolerant mutants

Conidial suspensions of *T. asperellum* were obtained from sporulating 3 days old culture by rinsing with sterile distilled water. After centrifugation at 5000 rpm for 10min, concentrated suspension of conidia was added to N-methyl-n-nitro-N- nitrosoguanidine (NTG) (100 mg/L in 0.05 M Tris buffer pH 6) for 1 h in a shaker (250 rpm/10m) at room temperature. The NTG treated conidial suspensions were washed thrice in sterile distilled water by centrifugation at 5000 rpm for 10min. Conidial suspension (0.1 ml) were plated on PDA containing 10 μg/ml carbendazim (97%, Sigma Aldrich), 500 mg/L streptomycin and 150 mg/L Rose Bengal. The plates were incubated at 25±1˚C with 12 h light/dark photoperiod and growth of the colonies were observed daily. After 5 days, colonies which were tolerating 10 μg/ml carbendazim (97%) were isolated and sub-cultured for 10 times on PDA without carbendazim to test their stability [19]. Stable carbendazim tolerant mutants were selected and designated as NTG (N1, N2 and N3) mutants. Further, these mutants were tested at different concentrations of carbendazim (10, 20...120 μg/ml). Mutants which showed the maximum tolerance to carbendazim (i.e., N2) was selected as the parental strain for $2^{nd}$ round of NTG treatment by following the above protocol. After $2^{nd}$ round of mutation three more carbendazim tolerating mutants were picked up and mutants referred as N2-1, N2-2 and N2-3.

## Screening and determination of sensitivity of the mutants

**Dual culture assay.** An actively growing mycelial disc (5 mm) of mutants and test pathogens (*F. oxysporum* f. sp. *ciceri*, *R. bataticola* and *B. cinerea*) were placed near the periphery of PDA amended Petri dishes and incubated at 25±1˚C with 12 h light/dark photoperiod. Inhibition of average radial growth of pathogen was calculated in relation to the growth on control [20]. Three replications were maintained for each treatment and Petri dishes containing test pathogen without *Trichoderma* served as control. Per cent inhibition of mycelial growth was measured by using below mentioned formula.

$$I = (C - T/C) \times 100$$

where I = Per cent inhibition of pathogen mycelium; C = Growth of pathogen in control; T = Growth of pathogen in Trichoderma inoculated plate.

**Sandwich systems for volatile interactions.** Sandwich plate technique as described by Li et al. [21] was used to test the volatile antifungal activity of mutants against the three chickpea pathogens. Actively growing 3 days old cultures (5 mm mycelial disc) of mutants and test pathogens were centrally inoculated in a separate PDA Petri dishes. Top of the Petri dishes were removed and base containing mutants and test pathogens were placed facing each other forming an atmosphere sharing set. Three replications were maintained for each treatment and incubated at 25±1˚C with 12 h light/dark photoperiod. Linear growth of test pathogens in inverted plates were measured when the test pathogens covered the Petri dish, and per cent mycelium inhibition was calculated by using above mentioned formula.

**Sensitivity of mutants to carbendazim.** Poisoned food technique was followed to study the carbendazim tolerance in mutants. To test the level of tolerance in 1st round NTG induced mutants, carbendazim at concentrations 10, 20, 30, 40, 50, 60, 70, 80, 90, 100, 110 and 120 μg/ml was supplemented with PDA. To test the level of tolerance in 2nd round NTG induced mutants, carbendazim at concentrations 50, 100, 150, 200, 250, 300, 350, 400, 450, 500, 600, 700, 800, 900, 1000, 1100, 1200, 1300, 1400 and 1500 μg/ml was supplemented with PDA. Each concentration of carbendazim was replicated four times and radial growth (mm) of mutants was measured after 7 days of incubation at 25±1˚C with 12 h light/dark photoperiod. The effective concentration ($EC_{50}$) values of 2nd round NTG induced mutants were calculated by measuring the radial growth mycelia against log value of carbendazim concentrations [22].

Maximum carbendazim tolerating mutants N2 and N2-2, along with wild-type (WT) were subjected for dry mycelia production at different concentration carbendazim. A 5 mm mycelial disc of these cultures were inoculated to potato dextrose broth (PDB) (HiMedia) containing carbendazim at 0, 250, 500, 750, 1000, 1250, 1500, 1750 and 2000 μg/ml. Flasks were incubated at 25±1˚C with 12 h light/dark photoperiod for 7 days without shaking. After 24 and 48 h of incubation, conidial germination and formation of mycelium in carbendazim amended flasks were observed under microscope (Olympus Corporation, Japan). After 7 days of incubation, cultures were filtered through Whatman (No.1) filter paper and mycelia was air-dried at 70˚C for 12h in hot-air oven. The dry weight (g) of the mycelium was recorded.

## Greenhouse experiments for management of chickpea dry root rot disease

**Development of talc-based formulations of mutants.** Based on carbendazim tolerance, mutants N2, N2-2 along with WT was selected for chickpea dry root rot disease management against moderately resistant Annigeri and JG 62 cultivars under greenhouse conditions. Actively growing 5 mm mycelial disc of these cultures were inoculated separately into 250 ml conical flask containing 100 ml PDB and incubated at 25±1˚C with 12 h light/dark photoperiod for 7 days without shaking. *Trichoderma* conidia load was fixed to $10^8$/ml and mixed with autoclaved talcum powder at 1:3 proportion (S1 Fig). Formulations were air-dried at room temperature and colony forming unit (CFU) was calculated by following serial dilution technique on TSM (Trichoderma-selective agar medium) containing 100 mg/L Rose Bengal.

**Bio-efficacy of mutants against chickpea dry root rot disease.** Disease suppression ability of mutants N2, N2-2 and WT was evaluated against chickpea dry root rot disease in completely randomized block design (CRBD). Dry root rot susceptible pigeonpea cultivar Annigeri and JG 62 seeds were treated with *Trichoderma* formulations at the rate of 5 g/kg seeds and half- recommended dose (RD) of carbendazim (Bavistin-WP 50) i.e. at the rate of 1 g/kg seeds. Treated seeds were shade dried before sowing. Chickpea dry root rot sick pots were prepared by following the protocol mentioned in ICRISAT Chickpea Diseases Phenotyping Bulletin (S2 Fig).

A total of eight treatments *viz.* $T_1$ (sick pot); $T_2$ (WT); $T_3$ (N2); $T_4$ (N2-2); $T_5$ (RD of carbendazim); $T_6$ (N2+0.5RD carbendazim); $T_7$ (N2-2+0.5RD carbendazim) and $T_8$ (Healthy pot) were levied separately on Annigeri and JG 62 cultivar. Seven seeds were sown per pot (13x8x9 cm) in 3 replications. Temperature (35˚C) of the greenhouse and moisture levels of the pots (60%) were carefully maintained as per the procedure given in the Bulletin. The percent disease incidence (PDI) was recorded at every four-day interval from the day of first disease appearance till complete wilting of all plants in the pots. The percentage of diseases incidence was calculated using the following formula.

$$\text{Per cent disease incidence} = \frac{\text{No.of wilt infected plants}}{\text{Total of plants assessed}} \text{X 100}$$

The dry root rot disease progress curves were constructed from the PDI data and the areas under the disease progress curve (AUDPC) and apparent infection rate (*r*) were calculated [23].

$$AUDPC = \sum_{i=1}^{n-1}[(X_{i+1} + X_i)/2][t_{i+1} - t_{1i}]$$

Apparent infection rate was estimated by regressing the logit of disease proportion (decimal from of per cent disease severity) on time in days [24].

## Assessment of translational changes in β-tubulin gene of mutants

**DNA extraction and amplifications of β-tubulin gene.** β-tubulin gene (*tub1* and *tub2*) was amplified from mutants N2, N2-2 and WT (S1 Table). The total genomic DNA was extracted from actively growing 3-days old mycelium using CTAB Method (Mukherjee et al. 2019) *tub1* and *tub2* genes were amplified in a 25 μl PCR reaction for 35 cycles (S2 Table). Amplified PCR product was electrophoresed in 1.5% agarose gel and bands were visualized under UV transilluminator.

PCR clean-up and gel extraction were followed by using Macherey-Nagel gel elution kit and was ligated with pJET1.2 blunt vector (Thermo Scientific Clone pJET PCR Cloning Kit). The ligated product was used to transform competent *E. Coli* DH5-α. Positive colonies were confirmed by using colony PCR. The plasmid was isolated using Macherey-Nagel plasmid DNA purification Kit, and vacuum air-dried. The clones were checked by Sanger sequencing (Macrogen Inc., South Korea).

## Sequencing and homology modelling of β-tubulin gene

All the obtained sequences were aligned and checked for sequence variation using Clustal omega (https://www.ebi.ac.uk/Tools/msa/clustalo/) and the alignment was visualised by using ESPRIPT 3.0 software (https://espript.ibcp.fr/ESPript/ESPript/) [25]. The β2-tubulin amino acid sequences were retrieved from the NCBI-Protein database in FASTA format and used as a reference for the alignment. The alignment was then analysed for occurrence of deleterious mutations, or amino acid substitution by using Provean (http://provean.jcvi.org/seq_submit.php) online tool. Homology modelling was carried out with the SWISS-Model interface with the β-tubulin X-ray crystallographic structure (PDB: 4ihj) as a template. All the models were evaluated by the QMEAN4 score. Global model quality estimation (Z score −1.22 and the QMEAN score 0.692) suggested that the modelled protein is of good quality. The homology models of mutated β-tubulin were built by altering the appropriate parent amino acids using the WHAT IF program (http://swift.cmbi.ru.nl/servers/html/index). The mutated models were subjected to overall energy minimization by steepest descent method, using GROMOS 43B1 force field. PyMOL (http://www.pymol.org/) was used for inspection of models and crystal structures while CastP server (http://sts.bioengr.uic.edu/castp/calculation.php) was used for pocket calculations using probe radius 1.4 Å.

## Statistical analysis

The data of antagonistic, volatile inhibition and abiotic stress (NaCl salt and temperature) tolerance of mutants were arcsine transformed to make residuals normal and then back-transformed for the presentation of the results [26]. The transformed data was subjected for analysis of variance (ANOVA) to know the level of significance at $p<0.01$ by using PROC GLM, SAS 9.4 (Statistical Analysis Systems Institute Inc. 2016). Significance of mean differences within mutants was tested by Duncan multiple range test at $p<0.01$ level of probability. Boxplots were generated to visualize the per cent inhibition of chickpea pathogens, salt and temperature tolerance in mutants using R statistical program (R Development Core Team

**Table 1. Growth and morphological characterization of NTG induced *Trichoderma* mutants.**

| Mutant | Mycelium growth (mm) | | | | |
|---|---|---|---|---|---|
| | 24h | 48h | 72h | 96h | Average* |
| N1 | 24.67 | 53.17 | 76.33 | 90.00 | 49.23e |
| N2 | 28.50 | 60.33 | 80.00 | 90.00 | 52.17cd |
| N3 | 22.17 | 53.17 | 80.00 | 90.00 | 49.47e |
| N2-1 | 28.83 | 60.67 | 79.67 | 90.00 | 52.23c |
| N2-2 | 26.50 | 60.17 | 80.33 | 90.00 | 51.8d |
| N2-3 | 32.83 | 67.50 | 79.67 | 90.00 | 54.4b |
| WT | 33.50 | 70.67 | 82.33 | 90.00 | 55.7a |

* Average followed by the same letters (in superscript) within a column are not significantly different (P < 0.01) according to Duncan's multiple range test.

2020). The EC$_{50}$ value of carbendazim tolerance of mutants were analysed by using "DRC" and "MULTCOMP" library in R statistical program.

## Results

### Evaluation of altered morphology of mutants

The NTG induced mutants were significantly differed from the wild-type in growth rate and most of the mutants covered the complete Petri plate within 96 h of incubation. Highest average mycelial growth was observed in WT (55.70mm) followed by N2-3 (54.40mm) and lowest growth was observed in N1 (49.23mm) and N3 (49.478mm) mutant respectively (Table 1). Mutants did not show any variation in morphology, sporulation, colony pigmentation, conidia and chlamydospores production level when compared to wild type.

### Screening of Trichoderma mutants for anti-fungal activity and volatile metabolites activity

The results showed that mutants significantly ($p<0.01$) caused reduction of chickpea pathogen mycelium. Inhibition of *F. oxysporum* f. sp. *ciceri*, *R. bataticola* and *B. cinerea* mycelium was more due to volatile metabolites as compared to antagonistic activity (S3 Fig). Maximum mycelial inhibition was observed in *F. oxysporum* f. sp. *ciceri* followed by *B. cinerea* and *R. bataticola* in both dual culture and sandwich assays with mutants (Table 2). In dual culture assays, mutant WT (69.81%) follwed by N1 (61.85%), N2-3 (60.12%), N2-1 (59.88%). N2

**Table 2. Evaluation of antagonistic and diffusible volatile compounds inhibition ability of NTG induced *Trichoderma* mutants against chickpea pathogens.**

| Mutant | Per cent mycelium inhibition | | | | | | | |
|---|---|---|---|---|---|---|---|---|
| | Dual cultural assay | | | | Sandwich plate technique | | | |
| | *F. oxysporum* f. sp. *Cicero* | *R. bataticola* | *B. cinerea* | Average* | *F. oxysporum* f. sp. *ciceri* | *R. bataticola* | *B. cinerea* | Average* |
| N1 | 71.11 | 53.70 | 60.74 | 61.85b | 63.33 | 53.70 | 60.00 | 59.01b |
| N2 | 67.41 | 54.81 | 55.56 | 59.26bc | 61.48 | 27.41 | 69.63 | 52.84c |
| N3 | 59.26 | 55.56 | 54.81 | 56.54c | 69.63 | 33.89 | 65.56 | 56.36c |
| N2-1 | 64.81 | 56.67 | 58.15 | 59.88bc | 70.00 | 41.11 | 72.22 | 61.11b |
| N2-2 | 67.78 | 55.19 | 50.74 | 57.9bc | 70.37 | 40.00 | 70.00 | 60.12b |
| N2-3 | 70.37 | 55.93 | 54.07 | 60.12bc | 78.52 | 42.96 | 72.59 | 64.69a |
| WT | 62.04 | 61.11 | 86.30 | 69.81a | 77.04 | 66.46 | 34.81 | 59.44b |

* Average followed by the same letters (in superscript) within a column are not significantly different (P < 0.01) according to Duncan's multiple range test.

(59.26%) and N2-2 (57.90%) showed the maximum inhibition of *F. oxysporum* f. sp. *ciceri*, *R. bataticola* and *B. cinerea* mycelium. Least inhibition of mycelium was recorded against mutant N3 (56.54%). On the other hand, mutants N2-3 (64.69%), N2-1 (61.11%), N2-2 (60.12%) and N1 (59.01%) showed the maximum volatile metabolites effects on *F. oxysporum* f. sp. *ciceri*, *R. bataticola* and *B. cinerea* mycelium inhibition. Least volatile metabolites effects were recorded in N2 (52.84%) and N3 (56.36%) mutants (Table 2).

## Evaluation of carbendazim tolerance and mycelial biomass production in mutants

After 2nd round of selection, it was noticed that mutants had surpassed natural tolerance to field dose of carbendazim (Bavistin-WP 50). The minimum inhibitory concentration (MIC) of the fungicide was calculated to determine the mutants' tolerance capacity. When the carbendazim tolerant mutants were grown at graded levels of carbendazim, mutant N2-2 and N2-3 was observed to be more compatible at high concentration of carbendazim (Fig 1).

The 6 NTG (3 each from 1st and 2nd round selection) treated mutants were developed. The mutants which were resistant to 10 µg/ml of carbendazim were tested at graded (10–120 µg/ml) levels of carbendazim. Mutant N2 showed maximum carbendazim tolerance compared to N3 and N1 mutants (Fig 2A). Further, N2 mutant was subjected for 2nd round of NTG irradiation and 3 more (N2-1, N2-2 and N2-3) mutants were recovered. At higher concentration, the growth of mycelium was negligible or almost nil in N2-1 as compared to N2-2 and N2-3 mutants (Fig 2B). The effective concentration ($EC_{50}$) values of these mutants were calculated where among the mutants, N2-2 (5626.40 µg/ml) showed the highest $EC_{50}$ values followed by N2-3 (206.36 µg/ml) and N2-1 (16.41 µg/ml).

Mycelial dry biomass production was significant at $p<0.01$ per cent in the mutants against different concentration of carbendazim. Maximum dry biomass production was noticed in N2-2 and results were at par with carbendazim at 0, 250, 500, 750, 1000 and 1250 µg/ml (Fig 2C). As the fungicide concentration increased, the mycelium biomass and conidial germination was also reduced at different concentration of carbendazim (S4 Fig).

## Disease control potentials of mutants

The disease control potential of previously observed carbendazim tolerating N2 and N2-2 mutants from 1st and 2nd round of selections were evaluated under greenhouse conditions with controls. The results obtained in the two separate experiments are shown in S3 Table. Mutant N2-2 along with half-dose carbendazim treatment ($T_7$) in both the cultivars, were registered with less disease incidence and higher plant growth compared to other treatments against chickpea dry root rot. Even after 60 days of sowing, adherence of *Trichoderma* mycelium was observed in the chickpea roots of $T_7$ treatment (Fig 3).

Treatment $T_7$ (N2-2+0.5 RD carbendazim) showed significantly lower AUDPC incidence (1980.95 and 1895.24) and apparent infection rate (0.11 and 0.11) in both the cultivars. Treatment $T_3$ (N2) and $T_4$ (N2-2) without half-dose carbendazim, showed the maximum AUDPC of incidence and apparent infection rate. There were no significant differences in case of disease progress and incidence between the cultivars Annigeri and JG 62 (Table 3).

## Sequence variations in the β-tubulin genes of carbendazim tolerance mutants

N2 and N2-2 mutants were analysed to determine the sites of mutation in β-tubulin genes. Alignment of a nucleic acid sequences of *tub1* and *tub2* genes in WT, N2 and N2-2 revealed that the mutations occurred only in *tub2* (protein length: 563 amino acid), whereas no mutations was found in *tub1* gene. The amino acid sequence demonstrated that there were over 211

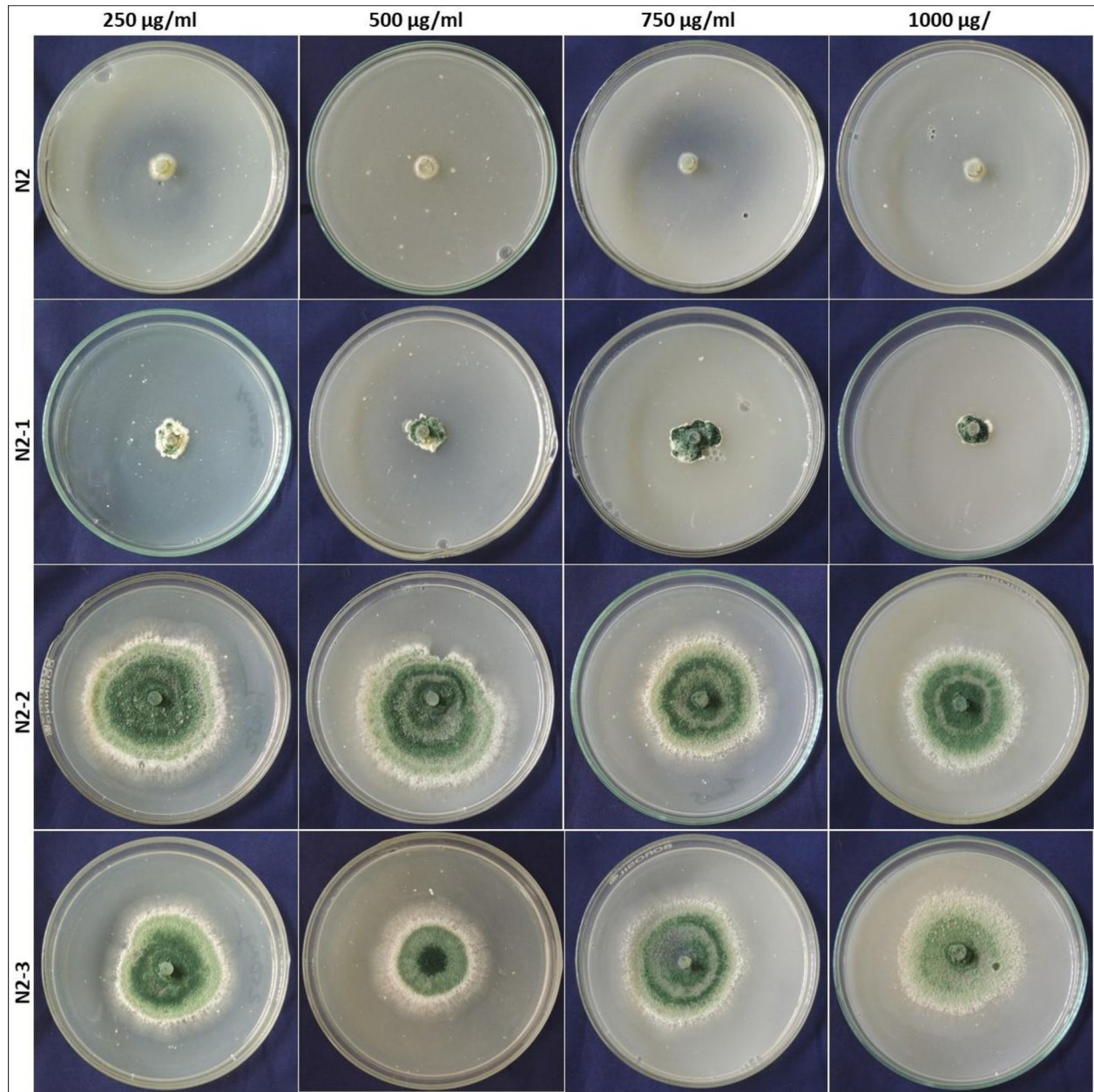

**Fig 1. Carbendazim tolerance in N-methyl-n-nitro-N-nitrosoguanidine irradiated *Trichoderma* mutants.** Mutants N2-1, N2-2, N2-3 is compared with N2 mutant at carbendazim 250, 500, 750 and 1000 µg/ml in PDA plates, and observation are recorded when the control plates cover the whole plate.

observed mutations (substitutions) in N2 and 275 in N2-2 out of which 17 and 14 were found to be deleterious, respectively (S4 Table). Substitutions occurring at positions 156, 157, 170, 171, 189, 190, 237, 238, 239, 247, 252, 253, 255, 277, 278, 279 and 362 in N2 were found to cause deleterious mutations which implied that the mutations were non-synonymous in nature.

Furthermore, two fragments between positions 176–177, and 286–297 were inserted while deletions from 248–251 and 300–306 were observed in N2. Similarly, in case of N2-2, substitutions at positions 169, 171, 247, 248, 249, 250, 251, 252, 253, 255, 277, 278, 279 and 342 caused

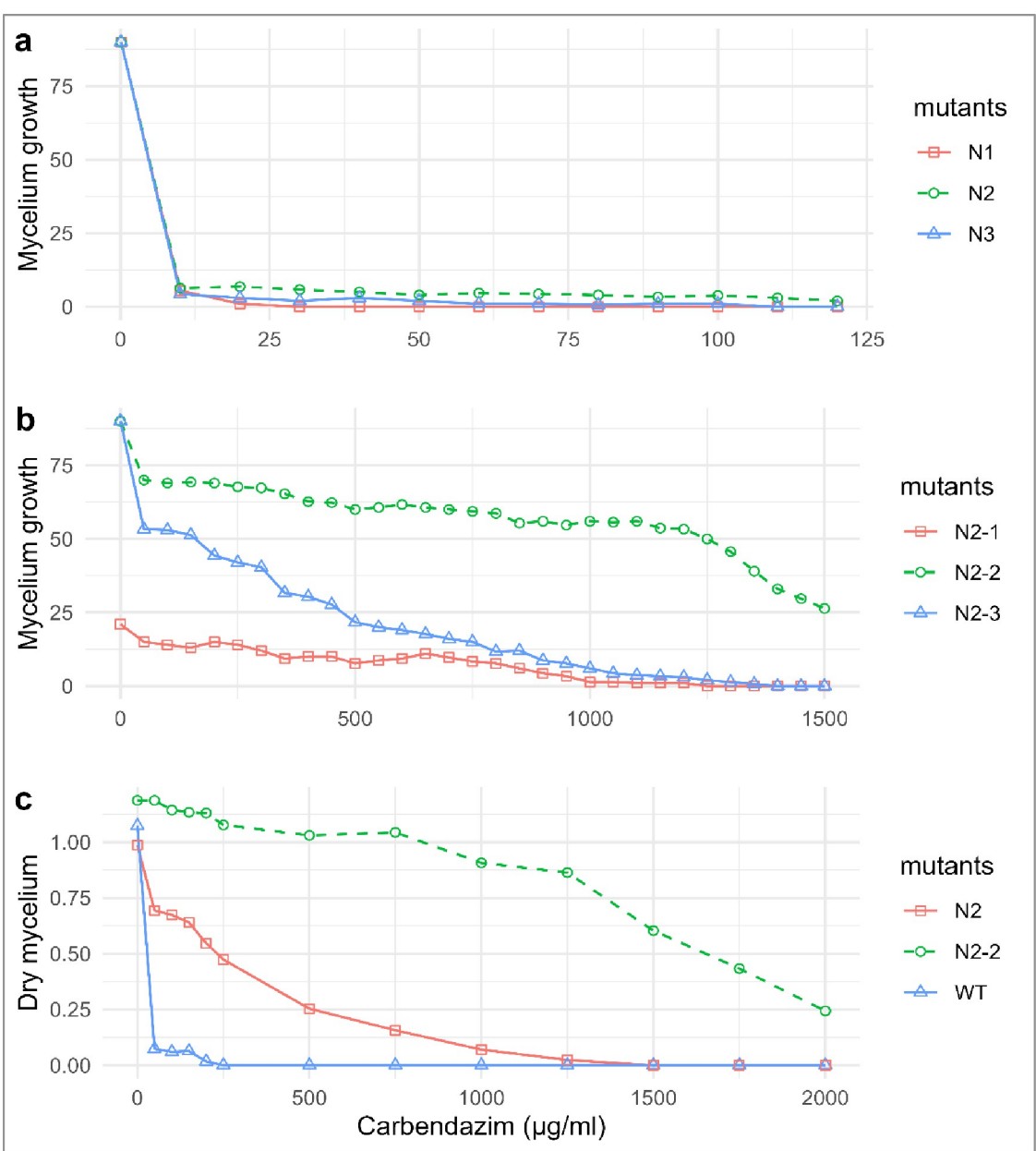

**Fig 2. Carbendazim tolerance and dry mycelial productions in N-methyl-n-nitro-N-nitrosoguanidine irradiated *Trichoderma* mutants. (a)** The 1st round NTG induced N1, N2, N3 mutants carbendazim tolerance at 0–120 µg/ml carbendazim amended PDA; **(b)** 2st round NTG induced N2-1, N2-2, N2-3 mutants carbendazim tolerance at 0–1500 µg/ml carbendazim amended PDA; **(c)** dry mycelium production ability of N2-2, N2 and WT in 0–2000 µg/ml carbendazim amended potato dextrose broth.

non-synonymous mutations. Fragments of insertions were found at 146–147, 338–340, 359–360 and deletions were found at 48–50, 293–300, 307–315 amino acid positions (S4 Table, S5 Fig). The deduced amino acids structural superimposition of N2 with WT [RMSD = 1.03, structure overlap = 96.51%] and of N2-2 with WT [RMSD = 1.20, structure overlap = 81.07%] gave 37 and 183 mutant residues (depicted as magenta) (Fig 4). This might be the result of synonymous/non-synonymous substitution of amino acids which would be responsible for mutation and ultimately carbendazim resistance.

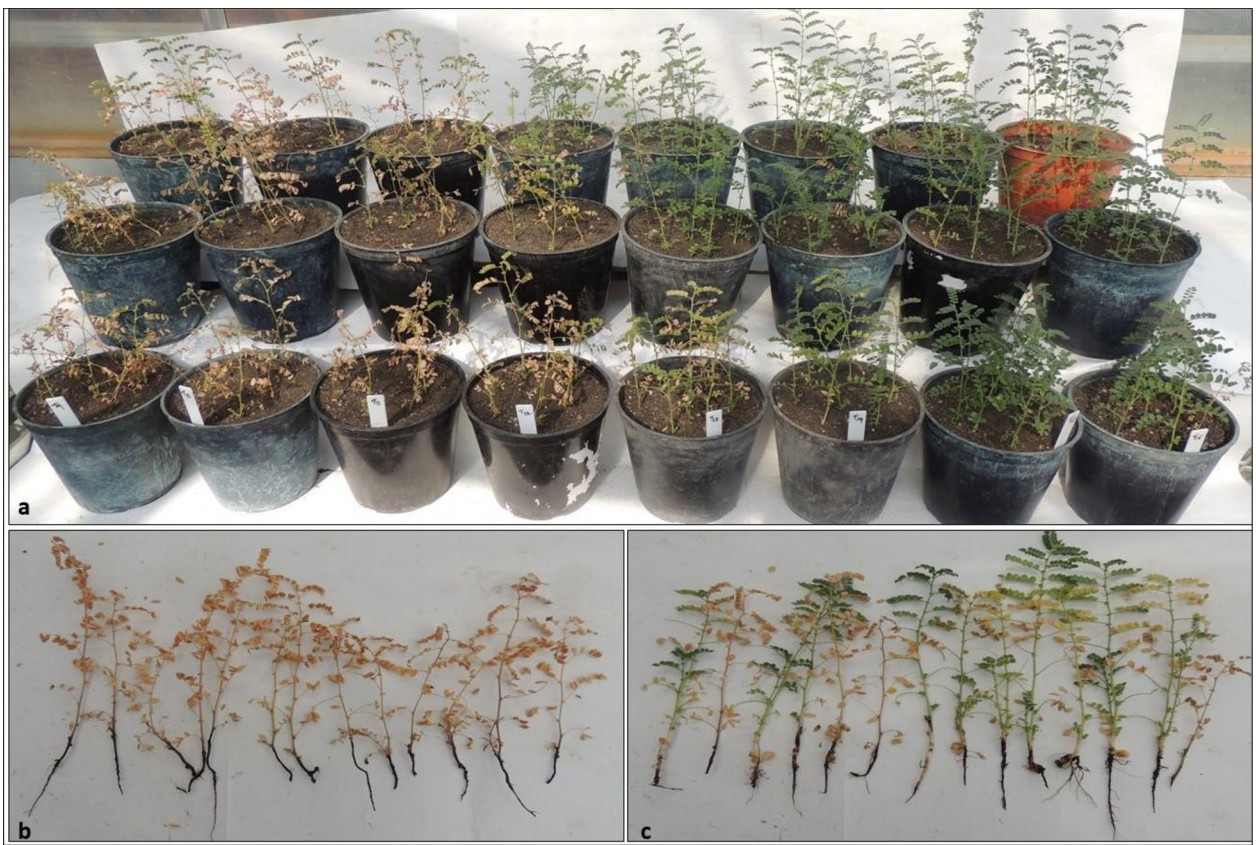

**Fig 3. Integrated management of chickpea dry root rot disease under glasshouse condition.** (a) Overview of the experiments at 44 days after sowing; (b) complete dry rot infected plants in $T_1$ (untreated) treatment; (c) partially dry rot infected plants in dry root rot in $T_7$ (N2-2+0.5 RD carbendazim) treatment at 60 days after sowing in JG 62 cultivar.

## Discussion

Biological control with improved *Trichoderma* strains has proven to be a potential alternative to chemicals of several soil-borne plant pathogens [27]. In many cases, individual use of either

**Table 3. Area under disease progress curve (AUDPC) and estimated apparent infection rate (*r*) in different treatments for chickpea dry root rot management under glasshouse conditions.**

| Treatment | Annigeri cultivar | | JG 62 cultivar | |
|---|---|---|---|---|
| | AUDPC (% x days) | Apparent infection rate | AUDPC (% x days) | Apparent infection rate |
| $T_1$ (sick pot) | 3504.76 | 0.24 | 3257.14 | 0.17 |
| $T_2$ (WT) | 3400.00 | 0.24 | 3219.05 | 0.21 |
| $T_3$ (N2) | 3542.86 | 0.30 | 3419.05 | 0.23 |
| $T_4$ (N2-2) | 3466.67 | 0.29 | 3228.57 | 0.18 |
| $T_5$ (RD of carbendazim) | 2961.90 | 0.17 | 2790.48 | 0.14 |
| $T_6$ (N2+0.5 RD carbendazim)) | 2638.10 | 0.13 | 2466.67 | 0.14 |
| $T_7$ (N2-2+0.5 RD carbendazim) | 1980.95 | 0.11 | 1895.24 | 0.11 |
| $T_8$ (healthy pot) | - | - | - | - |

The percent disease incidence of chickpea dry root rot was assessed at 20, 24, 28, 32, 36, 40, 44, 48, 52, 46, 60 days after sowing in sick pots. Date of the observations were used to calculate the area under the disease progress curve (AUDPC) (accumulative %); apparent infection rate (*r*) (unit progress/day).

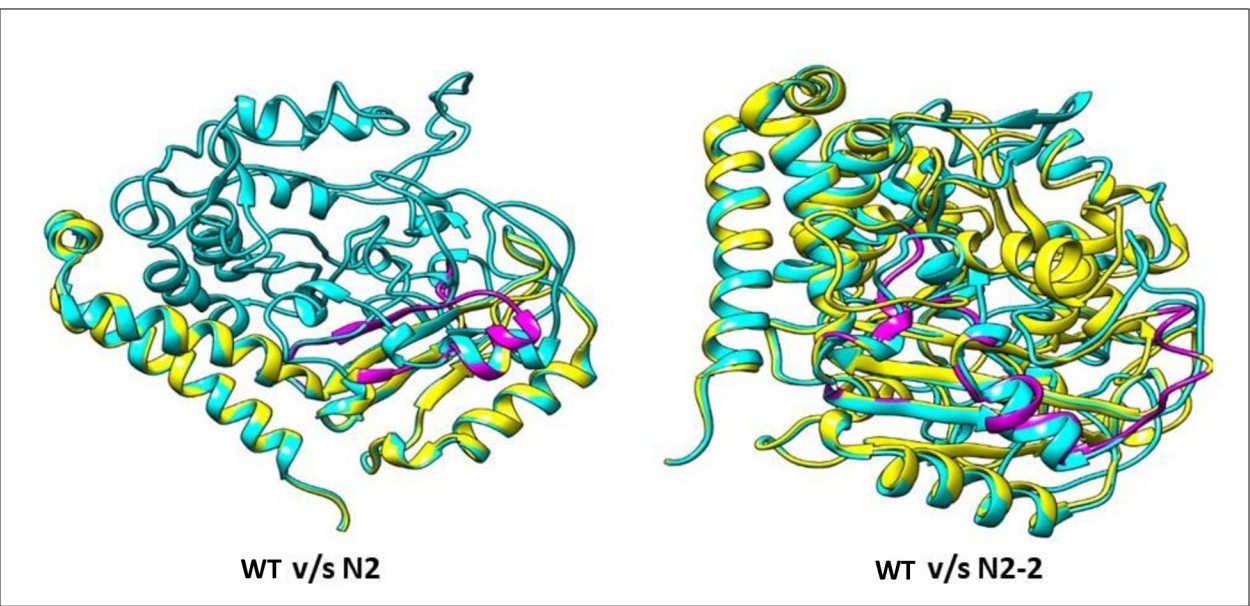

**Fig 4. Ribbon representations of *Trichoderma* N2, N2-2 and WT strain *tub2* protein.** Deduced amino acids structural superimposition between N2 and N2-2 with WT of tub2 gene differ by 37 and 183 mutant residues. In both the images—yellow (mutant protein), cyan (WT protein) and magenta (mutant residues).

biocontrol agents or chemicals could not provide satisfactory levels of control over soil-borne [8]. Attempts have been made to obtain resilient fungicide resistant strains for usage in integrated disease management [8]. The chemical irradiations followed by assessment of mutant's biocontrol efficacy have been guiding the selection of superior mutants for field application [1, 28]. In the light of enhancing biocontrol efficacy in *T. asperellum*, NTG induced mutants confirmed the fungicide tolerance which were morphologically similar to the wild-type [1]. In chemical mutagenesis, the recommended dose of mutagenic compounds should be as large as possible with minimum deleterious effects [29]. In our study, the optimal dose of NTG (100 mg/L) was sufficient enough to cause the mutations leading to carbendazim resistance. Repeated NTG induced mutation in carbendazim resistant mutant further increased the carbendazim tolerance. Possible reason for the difference in fungicide resistant among repeated mutants could be due to acclimatization to the shock events.

Production of cell wall degrading enzymes (CWDEs) and consortium of antifungal compounds are known to be the most important components of biocontrol by *Trichoderma* spp. [30]. Induced mutations may or may not increase the mycoparasitism activity of the *Trichoderma* spp. [31]. We found that mutants were capable of inhibiting mycelium of *F. oxysporum* f. sp. *ciceri* followed by *B. cinerea* and *R. bataticola* as compared to wild-type. This is in agreement with Abbasi et al. [28] who reported that *Trichoderma* mutants can increase inhibition of hyphal growth of *F. oxysporum*, *R. solani*, *R. bataticola*, *P. ultimum*, *M. phaseolina*, *S. sclerotiorum* and *F. grminearum* mycelium in both dual culture assays.

It was demonstrated that *Trichoderma* spp. usually use multiple biocontrol mechanisms, rather than a single mechanism to suppress plant pathogens [30]. The arsenal of volatile metabolites produced in *Trichoderma* spp. have shown its ability to inhibit the plant pathogens and promote plant growth [32]. We found that volatile metabolites produced by mutants were effectively inhibited the mycelial growth of *F. oxysporum* f. sp. *ciceri* followed by *B. cinerea* and *R. bataticola*. Reduction of mycelium growth in these pathogens are in accordance with Faheem et al. [33]. The study reported that volatile metabolites from *T. viride* (Tv-1) were

effective against *F. oxysporum*, *R. solani*, *S. rolfsii*, *S. sclerotiorum*, *C. capsici*, *H. oryzae*, *A. brassicicola* mycelium inhibition. It has been observed that *Trichoderma* spp. may produce specific volatile metabolites for each type of antagonized pathogen.

Quite often biocontrol agents fail to perform as expected against targeted crop disease. Combination of different disease management methods have significant impact on many perspectives. From practical point of view, combined application of biocontrol agents and half-dose of recommended fungicides give additive or synergistic effects on plant diseases [10] by weakening the pathogen propagules [34]. Carbendazim resistance in NTG induced mutants are stable. We found that linear growth and mycelial biomass production was more and highly compatible in N2-2 mutant. *T. harzianum* mutant developed by Viji et al. [35] showed carbendazim tolerance up to 30 µg/ml. Similarly, NTG induced *T. viride* mutants exhibited benomyl tolerance up to 50 µg/ml [19] and Jayaraj and Radhakrishnan [36] demonstrated 100 µg/ml tolerance in *T. harzianum*. Further, Mech [37] reported that mutants of *T. harzianum* can grow even at 0.1% carbendazim.

Combination of N2-2 with half-dose of carbendazim treatment ($T_7$) significantly reduced the dry root rot disease incidence in Annigeri and JG 62 cultivar. Since, these genotypes are known to be moderately resistant, the $T_7$ treatment has effectively enhanced the resistance capacity against DRR. JG62 is known to exhibit a mortality rate of 90% under DRR incidence placing them sometimes under highly susceptible genotypes as well [38]. AUDPC and apparent infection rate (*r*) indicated slow disease progress in this treatment. *R. bataticola* in legumes is becoming intense in tropical humid areas [39] moisture deficit conditions and higher soil temperatures which are predisposing factor for disease in chickpea [40]. Moreover, carbendazim tolerating *T. harzianum* mutants have been found superior in enhancing the root biomass of chickpea and increasing *F. oxysporum* f. sp. *ciceri* management to about four to eight fold [41].

In this study, we demonstrated that only *tub2* gene shows mutations in N2-2 and N2. The deduced amino acids structural superimposition of *tub2* gene in N2 with WT and N2-2 with WT, differ by 37 and 183 mutant residues. Carbendazim is known to inhibit the synthesis of β-tubulin protein [42]. Mutations in the β-tubulin gene evade the inhibitory effect of carbendazim and confer carbendazim tolerance to the mutant. A single amino acid change from phenylalanine to tyrosine at position 167 of the β-tubulin gene results in benomyl tolerance in *N. crassa* and *A. chrysogenum* [43]. Goldman et al. [15] confirmed the MBC resistant *tub2* gene from *T. viride* strain T9BR47. A single mutation at amino acid 168, having Phe (TTC) instead of Ser (TCC) and a double mutation in amino acid 13 resulting in the substitution of Gly (GGC) by Val (GTG) in *tub2* confirms the resistance in *T. harzanum*. Previous studies have demonstrated the mutations occurring at multiple codons of the β-tubulin gene ultimately confer tolerance to benzimidazoles such as carbendazim [8]. Our study reported 17 mutations in N2 and 14 mutations in N2-2 which were non-synonymous in nature.

This suggests that carbendazim tolerance could be attributed to mutations in *tub2* gene in N2 and N2-2. Although N2 and N2-2 mutants showed increased carbendazim resistance, their biocontrol properties that prevented dry root rot were not hampered. This suggests that the fungicidal and biocontrol characteristics of *Trichoderma* are not linked to naturally occurring *tub2* gene. This further shows that NTG induced mutagenesis allowed for selection of carbendazim tolerant *Trichoderma* mutants for management of dry root rot in chickpea. Further structural analysis of mutant *tub2* protein structures would be necessary to narrow down upon the causal mutation.

Our results showed that the fungicide tolerant *Trichoderma* strain with half-dose fungicide could control major chickpea fungal pathogens without affecting the original essential mycoparasitism and plant growth activity. These mutants were as good as the wild-type with respect to all of these inherent attributes.

## Conclusions

In conclusion, optimised mutagenic techniques improves the biocontrol potentials of *Trichoderma*. This study provides insight into the biocontrol potential of *T. asperellum* and its carbendazim tolerance. Additionally, the study demonstrates the soil-borne disease control ability of *Trichoderma* with half-dose of fungicide and confirms the carbendazim resistant mechanism in N2-2 mutants. The fungicide resistant *Trichoderma* mutants would certainly compliment the disease management practices with great environmental relevance.

To elucidate the antagonism and disease control potential of mutants with half-dose of fungicides against major chickpea pathogens, its molecular and biochemical basis of resistance needs to be examined. Whole genome sequences could explain the possible genetic causes for the heritably-stable changes in our *Trichoderma* mutant's antagonism and fungicide tolerance. There could be single and multiple point mutation for enhancing cell wall degrading enzymes and volatile metabolites productions and carbendazim-fungicide resistance in the mutants. Furthermore, experiments on tripartite interaction system that includes plant, plant pathogen and biocontrol agent (consortium of improved antagonistic and fungicide tolerant mutants) would provide the significance of improved biocontrol agents specifically in additive or synergistic effects on plant diseases, plant growth promotion, soil health as well as lowering chemical residue in crops and soil.

## Supporting information

**S1 Fig. Preparation of *Trichoderma* formulation and chickpea seed treatment.** (a) *Trichoderma* conidia suspensions at$10^8$/ml (a); (b) mixing of conidia suspension with talc powder at 1:3 proportion; (c) chickpea seed treatment with *Trichoderma* formulation and shade drying.
(DOCX)

**S2 Fig. Preparation of chickpea dry root rot sick pots under glasshouse condition.** (a) Actively growing *R. bataticola* mycelium in sand-maize meal medium; (b) mixing dry rot inoculum with autoclaved soil and growing on susceptible dry root rot cultivar to confirm the sickness of pots.
(DOCX)

**S3 Fig. Antagonistic and volatile metabolite effect of Trichoderma mutants on chickpea pathogen.** Per cent mycelium inhibition of chickpea *Fusarium oxysporum* f. sp. *ciceri*, *Rhizoctonia bataticola*, *Botrytis cinerea* in dual culture assay and sandwich systems for volatile metabolites effect.
(DOCX)

**S4 Fig. Conidial germination of N2-2 mutant at different concentrations of carbendazim.** (a) Carbendazim at 0 μg/ml after 24 h; (b & c) carbendazim at 1000 μg/ml after 24 and 48 h; (d & e) carbendazim at 1500 μg/ml after 24 and 48 h.
(DOCX)

**S5 Fig. Multiple sequence alignment of *Trichoderma asperellum* (N2, N2-2 and WT) *tub2* gene.** In the image TaTub2_mt1, TaTub2_mt2 and TaTub_WT -T5 represents the N2, N2-2 and WT strain.
(DOCX)

**S1 Table. Primer details of *tub1* and *tub2* for amplification of β-tubulin gene.**
(DOCX)

**S2 Table. PCR reaction and program.**
(DOCX)

**S3 Table. Integrated management of dry root rot of chickpea disease by carbendazim resistant *Trichoderma* mutants under glasshouse conditions at Annigeri and JG 62 cultivar.**
(DOCX)

**S4 Table.** Mutations occurring in a) N2 and b) N2-2 analyzed through Provean software. Insertions have been depicted as INS and deletions as DEL. Non-synonymous mutations through substitutions have been depicted as deleterious.
(DOCX)

## Acknowledgments

Authors thank Dr. P.K. Mukherjee, Bhabha Atomic Research Centre, Trombay, Mumbai, India for providing facilities for developing the mutants and guidance.

## Author Contributions

**Conceptualization:** Ramangouda G., M. K. Naik, Mamta Sharma.

**Data curation:** Rahul B. Nitnavare.

**Investigation:** Ramangouda G., Rahul B. Nitnavare.

**Methodology:** M. K. Naik, Mamta Sharma.

**Project administration:** Ramangouda G., M. K. Naik.

**Resources:** Pooja Bhatnagar-Mathur, Mamta Sharma.

**Software:** Rahul B. Nitnavare, Joorie Bhattacharya.

**Validation:** Richa Yeshvekar, Joorie Bhattacharya.

**Visualization:** Rahul B. Nitnavare, Richa Yeshvekar, Joorie Bhattacharya, Pooja Bhatnagar-Mathur.

**Writing – original draft:** Ramangouda G.

**Writing – review & editing:** Rahul B. Nitnavare, Joorie Bhattacharya, Mamta Sharma.

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
