## [Decision Letter · Decision Letter 0]

4 Nov 2022

PONE-D-22-28434Genetic Enhancement of Trichoderma asperellum Biocontrol Potentials and Carbendazim Tolerance for Chickpea Dry Root Rot Disease ManagementPLOS ONE

Dear Dr. Sharma,

Thank you for submitting your manuscript to PLOS ONE. After careful consideration, we feel that it has merit but does not fully meet PLOS ONE’s publication criteria as it currently stands. Therefore, we invite you to submit a revised version of the manuscript that addresses the points raised during the review process.

We look forward to receiving your revised manuscript.

Kind regards,

Abhay K. Pandey

Academic Editor

PLOS ONE

Journal Requirements:

"NO".

Reviewers' comments:

Reviewer's Responses to Questions

**Comments to the Author**

1. Is the manuscript technically sound, and do the data support the conclusions?

Reviewer #1: Yes

Reviewer #2: Yes

Reviewer #3: Yes

2. Has the statistical analysis been performed appropriately and rigorously? 

Reviewer #1: No

Reviewer #2: Yes

Reviewer #3: Yes

3. Have the authors made all data underlying the findings in their manuscript fully available?

Reviewer #1: Yes

Reviewer #2: Yes

Reviewer #3: Yes

4. Is the manuscript presented in an intelligible fashion and written in standard English?

Reviewer #1: No

Reviewer #2: Yes

Reviewer #3: Yes

5. Review Comments to the Author

Reviewer #1: Use more recent references

It’s better to know the Elicitors response rather than Beta Tubilin Gene against Mutant strains.

Colony of Trichoderma asperellum show different growth pattern from the growth pattern shown in fig 1 so authenticity of this isolate should be mentioned like Gene bank number etc

Study done is not similar with abstract. Please elaborate the abstract according to the results.

Reviewer #2: The paper is written in a detailed way and understandable for the researchers involved in biocontrol sector. the mutagenic techniques will helps in improving the biocontrol potential of Trichoderma Spp.

Reviewer #3: Kindly add the Gene bank AC to the all isolates in your study

Kindly check a word or phrase appears with and without a hyphen in comments and (anti fungal, bio agents soil borne, etc..)

You must complete the table of Abbreviations in attachment

Best wishes

6. PLOS authors have the option to publish the peer review history of their article (what does this mean?). If published, this will include your full peer review and any attached files.

Reviewer #1: No

Reviewer #2: **Yes: **Jeevan B

Reviewer #3: **Yes: **Prof. Khaled Hussein Arafat

---

## [Author Response · Author response to Decision Letter 0]

7 Dec 2022

The authors thank the reviewers for their valuable inputs which has enhanced the quality of our manuscript. The suggested changes have been duly made to the best of our knowledge.

Reviewer 1

Use more recent references

Thank you for the comment, as suggest new suitable reference were added

It’s better to know the Elicitors response rather than Beta Tubilin Gene against Mutant strains.

The authors thank the reviewer for rightfully pointing this out. we are planning a next set of experiments to understand the elicitors responses 

Colony of Trichoderma asperellum show different growth pattern from the growth pattern shown in fig 1 so authenticity of this isolate should be mentioned like Gene bank number etc

As suggested, GenBank accession number mentioned in the materials and methods sections

Study done is not similar with abstract. Please elaborate the abstract according to the results.

Thank you for the comment, as suggested changes are made in the abstract in accord to results

Reviewer 2

The present manuscript is comprehensive and imparts a new insight into biological control of plant diseases. However, there are few revisions as follows. 

L. 85 (DRR → (DRR)

Changed (L. 88)

L. 91 Trichoderma asperellum (WT) strain used in the study was obtained from UAS Raichur → Mention the accession number if any 

Added (L. 93)

L. 125 – 126 Per cent inhibition of mycelial growth was measured → Mention the Formula 

Formula mentioned (L.131-135)

L. 135 – 136 per cent mycelium inhibition was calculated → Using the formula….

Changed the sentence (L. 145)

L. 171 Cultivar Annigeri and JG 62 → Why the authors selected these cultivars for this study, whether they are susceptible for all the three diseases?? If so, indicate that, they are susceptible cultivars for their respective disease. 

Changed the sentence (L.180-181)

L. 195 – 196 → Mention the PCR conditions for the amplification of tub1 and tub2 genes.

It is mentioned in the S2 table (L. 207)

Reviewer 3

L.1 fungicide-resistance →fungicide resistance

L. 36 dry-mycelial →dry mycelial

L. 36 added GenBank accession

L. 93 we don’t have GenBank accession of chickpea pathogens 

L. 392 carbendazim-tolerance→ carbendazim tolerance

L. 427 volatile-metabolites→ volatile metabolites

L. 430 fungicide-tolerant→ fungicide tolerant

Table 1 changed 

Table 2 changed 

Mentioned the abbreviation accept standard once

---

## [Editor Report · Decision Letter 1]

20 Dec 2022

Genetic enhancement of Trichoderma asperellum biocontrol potentials and carbendazim tolerance for chickpea dry root rot disease management

PONE-D-22-28434R1

Dear Dr. Sharma,

We’re pleased to inform you that your manuscript has been judged scientifically suitable for publication and will be formally accepted for publication once it meets all outstanding technical requirements.

Kind regards,

Abhay K. Pandey

Academic Editor

PLOS ONE

Additional Editor Comments (optional):

authors addressed all comments
---

## [Editor Report · Acceptance letter]

6 Jan 2023

PONE-D-22-28434R1 

Genetic enhancement of *Trichoderma asperellum* biocontrol potentials and carbendazim tolerance for chickpea dry root rot disease management 

Dear Dr. Sharma:

I'm pleased to inform you that your manuscript has been deemed suitable for publication in PLOS ONE. Congratulations! Your manuscript is now with our production department. 

Kind regards, 

on behalf of

Dr. Abhay K. Pandey 

Academic Editor

PLOS ONE